# Analysis of Gene Differences Between F and B Epidemic Lineages of Bandavirus Dabieense

**DOI:** 10.3390/microorganisms13020292

**Published:** 2025-01-28

**Authors:** Wenzhou Ma, Yujia Hao, Chengcheng Peng, Duo Zhang, Yuge Yuan, Pengpeng Xiao, Nan Li

**Affiliations:** 1Wenzhou Key Laboratory for Virology and Immunology, Institute of Virology, Wenzhou University, Wenzhou 325035, China; ma1172864201@outlook.com (W.M.); unih4519@163.com (Y.H.); 18741083225@163.com (C.P.); zd1251071348@163.com (D.Z.); yuge202210@163.com (Y.Y.); 2College of Veterinary Medicine, Jilin University, Changchun 130062, China

**Keywords:** bandavirus dabieense, lineages division, mutation analysis, protein functional sites, propagation analysis

## Abstract

The prevalence of SFTS is becoming increasingly widespread and is expected to become a significant security issue. The article discusses the prevalence regions and genetic differences in two SFTSV lineages, so as to provide a scientific data basis for the clinical control and prevention of fever with thrombocytopenia syndrome. The literature involving SFTSV patients from 2009 to 2023 and SFTSV complete genome sequences uploaded by NCBI were collected and sorted out, based on time and SFTSV lineage division, we analyzed viral gene sequence. SFTSV patient data were continuously reported from 2009 to 2023, involving five countries including China, South Korea, Japan, Thailand, and Vietnam. There are obvious lineage and host divisions between the SFTSV lineages prevalent in China and abroad. The sources of B-lineage SFTSV samples are mainly concentrated in South Korea, Japan, and the middle and lower reaches of Hubei or Zhejiang in China, with half of the samples coming from humans and half from animals, and the F series SFTSV samples were mainly collected from provinces such as Anhui and Henan in China, with the main source being human patients. The F-lineage SFTSV is the highest proportion in the middle and upper provinces in China. The B lineage has recently appeared in Zhejiang and Taiwan and is prevalent abroad. Using prediction software based on molecular structure prediction technology, analyze the differences between the B and F lineages of SFTSV through prediction methods such as nucleotide mutations, gene recombination, mutation sites, and evolution rates. Conclusively, the differences in SFTSV between B and F lineages may be related to gene recombination of M and L fragments, it was also found that the B lineage had a lower recombination rate and mutation rate than the F lineage, and the evolutionary rate was prominently different. Comparative analysis of the differences in two SFTSV lineage genes could further understand the epidemic status of SFTSV and provide help and more insights for the prevention of the spread of specific types of SFTSV.

## 1. Introduction

*Bandavirus dabieense* is a newly emerged tick-borne virus, also known as SFTSV (Severe Fever with Thrombocytopenia Syndrome Virus) and HYSV (Huaiyangshan Virus) [1,2]. The original documented discovery of SFTSV was in 2009 [3], and the earliest patient report was in 2010. SFTSV was first isolated from patients in Hubei and Henan provinces in China in 2011. In the following years, South Korea [4,5], Japan [4,5], Thailand [6], Vietnam [6,7], Myanmar [6], and Pakistan [8] also reported cases of SFTSV. The number of countries where SFTSV is prevalent has increased and the scope of prevalence began to expand. SFTSV was first discovered to be transmitted through the bites of ticks, among which *Haemaphysalis Iongicornis* was the main one. Later, it was discovered that SFTSV has multiple modes of transmission. Infected animals, humans, and their secretions can all become sources of transmission, even aerosols [2,3].

SFTSV consists of three independent gene fragments, namely S (small), M (medium), and L (large) single-stranded RNA fragments. The S fragment encodes Np (Nuclear Proteins) and NSs (Nonstructural Proteins). The M fragment encodes glycoproteins GP (Glycoprotein Precursor). The L fragment participates in encoding RNA polymerase RdRp (RNA-dependent RNA polymerase) [9]. Each fragment can be separately classified into lineages, currently, there are multiple classification models for SFTSV. The mainstream classification divides the three fragments into six lineages—A, B, C, D, E, and F—but there are significant classification issues among them. For example, there is a significant cross-use phenomenon in the naming of the three lineages of F, E, and B. Some F lineages are referred to as E lineages, while others are referred to as F lineages [10,11,12,13,14,15]. Their division criteria may be the same, but there is a problem of inconsistent nomenclature. However, there is a common feature among the SFTSV lineages with different classification methods: one of the lineages always comes from multiple countries (B lineage). It is speculated that this lineage may have unique characteristics that distinguish it from other lineages, so the lineage (F lineage) with the highest number of uploads is selected for comparative analysis. This article reconstructed a full genotype SFTSV evolutionary tree with all uploaded data and based on the classification criteria of the previous literature, selected sequences from the B and F lineages for gene analysis.

## 2. Methods

### 2.1. SFTSV Epidemic Timeline

Through PubMed (https://pubmed.ncbi.nlm.nih.gov/) and CNKI (https://www.cnki.net/), we searched for most of the SFTSV data and literature and registered the time of prevalence, country, and source of virus samples of SFTSV that were once prevalent based on the literature content. We screened relevant information to build a statistical table (which excluded some individual cases) classified and summarized it mainly based on time, host, and country, and built an SFTSV epidemic timeline based on the table content. Through the analysis of SFTSV epidemic time, we can understand the prevalence of SFTSV.

### 2.2. SFTSV Lineages Analysis

Since its discovery, SFTSV has only been around for over a decade, so there is no clear standard for the division of SFTSV lineages. In order to explore the lineage division of various countries during the SFTSV transmission process, all complete genome sequences of SFTSV were downloaded from NCBI (https://www.ncbi.nlm.nih.gov/). All uploaded sequences from Japan were marked as incomplete, so only gene sequences with a completeness greater than 90% were selected as complete sequences for analysis. The genome sequences were screened to remove sequences with no country, no host, and no collection time. The specific SFTSV lineages in the literature were added to construct an ML tree to divide the lineages. Clarifying the division of lineages can better understand the prevalence of SFTSV in each country and the prevalent lineages. Here, due to the fact that only 50 sequences in the B lineage of the S segment of SFTSV can be analyzed, subsequent analyses of the F and B lineages will be screened based on 50 sequences. Before conducting sequence analysis, 100% identical sequences will be removed. Then, using animal hosts as the first screening principle and year differences as the second screening principle, while ensuring the existence of these two foundations, random selection of sequences is carried out to around 50 sequences for subsequent analysis.

In total, 1176 S sequences, including 1023 sequences from China, 125 sequences from Japan, 25 sequences from South Korea, and 3 sequences from Thailand were downloaded. It can also be divided into 1115 human sequences and 72 animal sequences (including 18 ticks). After filtering by conditions adding sequences with known lineages, and ultimately obtaining 1347 S sequences to construct an ML tree. We re-examined the virus here and found that some sequences were uploaded at the family level or only belonged to other viruses of the same genus, which were removed again. After screening, 970 S fragments were finally classified into lineages, 952 human sequences, 18 animal sequences, and 1094 M sequences, including 937 sequences from China, 133 sequences from Japan, and 24 sequences from South Korea. It can also be divided into 985 human sequences and 96 animal sequences (animal sequences including 18 ticks, and 13 sequences without host information). Adding sequences with a giving lineage, and finally obtaining 1233 M sequences to construct an ML tree, we re-examined the virus here and found that some sequences were uploaded at the family level or only belonged to other viruses of the same genus, which were removed again. After screening, 1094 M fragments were finally classified into lineages, 985 human sequences, 96 animal sequences, 13 blank host, and 1044 L sequences, 906 from China, 114 from Japan, and 24 from South Korea. It can also be divided into 964 human sequences and 74 animal sequences (including 21 tick sequences), adding sequences with selecting lineage, and ultimately obtaining 1184 sequences to construct an ML tree. After screening, 962 L fragments were finally classified into lineages, 911 human sequences, 45 animal sequences, and 13 blank hosts. (All animal samples include Cattle, Goat, Procyon lotor, Dog, Rat, *Haemaphysalis longicornis*, *Erinaceus amurensis*, *Felis catus*, *Procyon lotor*, *Amblyomma testudinarium*, etc. For more detailed data, please refer to Appendix A).

### 2.3. Nucleotide and Amino Acid Mutations in SFTSV

All the sequences of the lineage division were used for statistical analysis of SFTSV nucleotide and amino acid mutation sites. The S, M and L sequences of SFTSV with the earliest upload time to NCBI were selected as templates for comparison using tools such as Excel, Bioedit 7.2.5 [16], and BioAider_v1.527 [17]. The nucleotide and amino acid mutation rates of SFTSV were analyzed as a whole, and meaningful mutations and mutations with changes in amino acid properties were analyzed. Based on the data conclusions of the lineage division above, the nucleotide mutations of the F and B lineage were analyzed separately, and a 3D visualization model simulating the amino acid mutations of mutant proteins was constructed to explore the changes in the properties of proteins after mutation.

### 2.4. Protein Functional Sites Prediction of SFTSV

A part of F and B lineages were randomly selected from the SFTSV to predict the protein functional sites of SFTSV amino acid sequences. The palmitoylation of protein sequences was predicted using CSS-Plam2.0 [18] software. SUMOylation sites were predicted using the online website (https://sumo.biocuckoo.cn/index.php, accessed on 28 June 2024). N-glycosylation sites of protein sequences were predicted using the website (https://services.healthtech.dtu.dk/services/NetNGlyc-1.0/, accessed on 28 June 2024).

### 2.5. Genetic Recombination

The recombination analysis of SFTSV was performed with all lineages and the possibility of gene recombination of SFTSV was analyzed by using RDP4 4.830 and SimPlot_v3_5_1. RDP4 [19] and SimPlot [20] were used to verify each other. It imported the three SFTSV fragments into RDP4 for gene recombination analysis, with a *p*-value set no greater than 0.05, and selected RDP, Chimaera BootScan, GENECONV, MaxChi, SiScan, 3Seq as analysis method, tacitly approved other options. We summarized the sequence obtained by RDP4 and imported it into SIMPLOT, using the earliest sequence as a reference to visualize whether there are recombination breakpoints in the SFTSV fragments.

### 2.6. Evolution Rate

Based on the results of the SFTSV pedigree division, about 50 SFTSV sequence pedigrees were screened for evolutionary rate analysis in the time dimension. Specific operation steps: After comparing the sequences of two lineages by using MAFFT version 7 (https://mafft.cbrc.jp/alignment/server/index.html, accessed on 28 June 2024), construct an ML tree with IQTREE 1.6.12, and observe the time signal of the ML tree by TempEstv1.5.3 to confirm that the time signal is greater than 0. Use the PhyloSuitev1.2.3 to search for the best BEAST model for aligning sequences, then set BEAST parameters, filter BEAST model parameters later, and finally select the parameters. An MCC tree was constructed to compare two lineages. The MCC tree is finally obtained while ensuring convergence values, and Figtree v1.4.4 is used to beautify it. The results of the MCC tree are described and analyzed for the propagation path of data with relatively high credibility using spreaD3-v0.9.7.1rc.jar.

## 3. Result

### 3.1. SFTSV Continuous Epidemic Timeline

Since SFTSV was discovered in China, it has not only become prevalent in some provinces in China, but patients and animals carrying SFTSV have also gradually been found in other countries. By collecting the SFTSV literature to establish an SFTSV epidemic timeline (Appendix A), and constructing an SFTSV epidemic timeline (Figure 1), we can realize the prevalence of SFTSV: from 2009 to 2023, there have been reports of SFTSV cases every year, and these reports are mainly concentrated in several countries in East Asia, including China, Japan, South Korea, Thailand, and Vietnam. China was the first country to detect SFTSV. From the discovery of SFTSV in 2009 to 2023, there were literature reports of SFTSV patients in many provinces in China, especially in some high-incidence provinces such as Shandong Province, Jiangsu Province, Anhui Province, Henan Province, Hubei Province, Zhejiang province, etc. There were also some animals carrying SFTSV in the above provinces, mainly livestock, or some wild animals. In South Korea and Japan, the first SFTSV patients were discovered in 2013. Over the next 6 years, there have been continuous reports of SFTSV patients in South Korea. Regarding animal inspection, South Korea only conducted SFTSV tests on animal samples in 2013, 2015–2021, and mainly detected the presence of SFTSV in wild animals and *Haemaphysalis Iongicornis*. SFTSV patients were discovered in Japan from 2013 to 2020. Compared with China and South Korea, Japan had relatively more positive animal samples. In recent years, Vietnam and Thailand began to discover SFTSV patients in 2017 and 2019, respectively, but none of the experimental literature has explored the detection of plenty of positive animal samples.

### 3.2. SFTSV Lineage Analysis

The S fragment includes A, B, D, E, and F lineage, without C lineage. The F lineage had the largest number of sequences, with a total of 678 (50.3%) sequences, followed by lineages A, D, and B, with 227 (16.8%), 195 (14.4%), and 226 (16.7%) sequences, respectively. The E lineage had the least, with only 21 (1.5%) sequences. The M segment includes all six lineages, with the F lineage having the largest number of sequences, with a total of 563 (46.4%) sequences, followed by the A, B, and D lineages, including 238 (19.6%), 207 (17%), and 174 (14.3%) sequences, and the least are E and C lineages, with only 22 (1.8%) and 9 (0.7%) sequences; fragment L also contains all lineages, and the largest number of sequences is the F lineage, with a total of 557 (47.0%) sequences, followed by the A, B, and D lineages, including 224 (18.9%), 192 (16.2%), and 182 (15.3%) sequences. The least are the E and C lineages, with 18 (1.5%) and 10 (0.8%) sequences. All data were counted to construct a lineage division table (Appendix A). From the above results, it can be seen that the C lineage division of the S fragment is still unclear, while the three fragments of SFTSV have similar quantity distributions in the lineage division.

To explore the prevalence of SFTSV lineages in different countries, through phylogenetic analysis at the genetic level, Figure 2 is obtained. It clearly showed that most of the data on the S, M, and L segments came from China, and a small part came from South Korea, Japan, and Thailand. Currently, there is a prevalence of various lineages of SFTSV in China, especially in Hunan province, which has always been a high-risk area for SFTSV, and various lineages of SFTSV can be detected. The most SFTSV lineage in the data uploaded by China is the F lineage, accounting for about 46.4~50.3% of all uploaded complete genome lineages. However, in the SFTSV data uploaded abroad, the number of animal and human blood samples is roughly the same. Most of the isolated SFTSV belong to the B lineage, and only a few belong to other lineages. For this reason, the F and B lineages will be analyzed next.

The F lineage of three segments S, M, and L of SFTSV were concentrated in the central and eastern regions of China, such as Henan, Hebei, Hubei, and Anhui provinces, with a total time span of 2010 to 2022. As for other SFTSV F lineage viruses, only a few of them appeared in South Korea in 2012. During 2019, the S and M segments detected in cats in Japan were classified as F lineage. The animal host sources of F lineage included cats, mice, and *Haemaphysalis Iongicornis*. Considering the total SFTSV lineage from animals, F lineage is quite rare. The F lineage occupied the vast majority and was far ahead in the uploaded data of the three segments of SFTSV. From this, it can be inferred that the severe fever with thrombocytopenia syndrome in China is mainly dominated by the F lineage (without considering the recombination between the segments). Compared with the F lineage, the B-lineage SFTSV spread more to the southeastern region of China. In terms of the basic spread range, new areas such as Zhejiang and Taiwan have appeared. Moreover, most of the B-lineage SFTSV was uploaded by South Korea and Japan (Thailand only uploaded 3 S sequences, all of which belonged to the B lineage). A separate search of all SFTSV animal samples found that most of them also belonged to the B lineage. It is speculated that the B-lineage SFTSV may be more likely to infect animals or be more easily detected in animals. To explore the differences between the F lineage and the B lineage, the following genetic analysis of the two lineages was performed.

### 3.3. Nucleotide Mutation of SFTSV

The nucleotide and amino acid mutation analysis of the above data of SFTSV S, M, and L was constructed (Figure 3). As can be seen from Figure 3A, the nucleotide mutation levels of M and L fragments were relatively higher than S fragments, but there was no significant difference in nucleotide mutation levels among the three fragments. Thereafter, it found that the amino acid mutation frequency was relatively high in the M fragments after analyzing amino acid mutation frequency results, and a large number of amino acid mutations had occurred, while the S and L fragments were relatively conservative. Overall, the mutation frequency of the S and L segments was lower than that of the M segments. Then, a separate analysis of the F and B lineage was conducted to construct amino acid and nucleotide mutation analysis graphs for SFTSV F and B (Figure 3B,C). From the graph, it can be observed that the B-lineage SFTSV had fewer mutations in the M and L segments compared to the F lineage, with the most significant one being that all nucleotide mutations in the S segment did not alter the polarity of amino acids. It was speculated that this may be one of the potential reasons why the B lineage was prone to infecting animals and spreading widely abroad.

F and B lineages statistical analysis was performed on mutation data. A 3D protein model (Figure 4) was constructed using SWISS simulation based on initial sequence and amino acid characteristics. Visualizing some amino acid mutation sites, it was found that both the F and B lineage of SFTSV had their own unique amino acid mutations and shared mutation sites. The S segment of the B lineage did not show any changes in amino acid properties. Amino acids with meaningful mutation differences between the two lineages were marked. The M and L segment statistical data showed that the M segment F lineage had 141 unique amino acid property mutations. The B lineage had 59 unique amino acid property mutation sites, and the two lineages had 38 common amino acid mutations. There were 137 unique amino acid property mutations in the F lineage of L fragment, and 67 unique amino acid property mutation sites in the B lineage, for 28 common amino acid mutations.

### 3.4. Prediction of Protein Functional Sites for SFTSV F and B Lineages

The protein functional sites of SFTSV were predicted based on three fragments of SFTSV (Figure 5A). The selected SFTSV lineage had not detected SUMOylation sites. From the summary of the predicted results (Figure 5B), we can understand the palmitoylation situation of the two SFTSV lineages. There were differences in the SFTSV prediction results between the F and B lineages. Both lineages of the S fragment did not have palmitoylation sites, while the F lineage of the M fragment had nine palmitoylation sites, including 12, 17, 374, 376, 576, 916, 918, 919, and 921 (the numbers represent the predicted sequence of amino acid residues, which are synonymous in the following text). The B lineage had 18 palmitoylation sites, including 1, 6, 9, 12, 14, 15, 17, 365, 373, 375, 376, 576, 907, 915, 917, 918, 920, and 921. The F lineage L fragment only had one palmitoylation site located at position 7 and the B lineage has two palmitoylation sites located at 2 and 7. The predicted results of N-glycosylation are shown in Figure 5C. The F and B lineage of the S fragment had the same N-glycosylation site located at position 17. The F lineage of the M fragment had three N-glycosylation sites, including 33, 63, and 853. And the B lineage had six N-glycosylation sites, including 22, 30, 33, 52, 60, and 63. The F lineage L fragment had five N-glycosylation sites, including 503, 509, 853, 981, and 1834. The B lineage had seven N-glycosylation sites, including 341, 498, 503, 509, 853, 981, and 1834. These findings suggested that there were differences in protein modification sites between the F and B lineage of the SFTSV lineage, but not very obvious ones. This suggested that there may be a certain connection between lineage and protein function, but further research is needed to clarify the functional significance of these modifications on different lineages of SFTSV.

### 3.5. Genetic Recombination of SFTSV

Gene recombination analysis. Different numbers of recombinant sequences were obtained in three segments (Appendix A). It was found that 10 sequences may undergo recombination in the S segment, 20 sequences in the M segment, and 24 sequences in the L segment. Use the same gene recombination software SimPlot-v3_5_1 for prediction and validation again (Appendix A). Finally, it is predicted that there are 3 recombinant fragments in the S segment, 11 recombinant fragments in the M segment, and 24 recombinant fragments in the L segment. Genealogy reference was performed on the sequences that underwent recombination, and it was found that there were three B-lineage fragments in the M fragment, namely OQ388971.1 (B), MG920820.1 (B), and OM452787.1 (B). The L fragment contained both B-lineage and F-lineage recombination fragments, with B-lineage fragments being AB983501.1 (B) and KR698352.1 (B), and F-lineage fragments being MT320802.1 (F), MT320802.1 (F), OM453577.1 (F), OM453580.1 (F), OM453579.1 (F), OM453562.1 (F), OM453599.1 (F), and OM452967.1 (F). OM453583.1 (F). This significant difference suggests that the differences between the F and B lineages may be related to a single fragment in SFTSV.

### 3.6. Evolution Rate of SFTSV F and B Lineages

In total, 50 S sequences, 53 M sequences, and 53 L sequences were selected from the B lineage of SFTSV. In total, 60 S sequences, 44 M sequences, and 57 L sequences were selected from the F lineage for evolutionary rate analysis. Sequence selection eliminated the possibility of gene recombination and ensured relatively complete sequence information. According to the BEAST results, it was known that statistical analysis was still needed for the data of various analysis models to determine the optimal model (Appendix A). After the screening, the most suitable BEAST parameter setting for the three segments was GTR+All+GI+4+Uncorrelated relaxed clock+Bayesian skyline. After performing BEAST analysis and obtaining available convergence values (ESS > 200), the evolutionary rates and MCC trees (Figure 6) of the three segments of the two lineages were obtained. (The MCC tree of the failed data construction is shown in Appendix A).

The earliest ancestors of the SFTSV F and B lineage were predicted via the MCC tree. The evolution rate of the B-lineage S segment was 3.21 × 10^−4^, and the earliest ancestor prediction time was about 1953. The evolution rate of the F-lineage S fragment was 2.43 × 10^−4^, and the earliest ancestral prediction was around 1984. The evolution rate of the B lineage M segment is 3.21 × 10^−4^, and the earliest ancestor prediction time was about 1953. The evolution rate of the F lineage M segment is 2.43 × 10^−4^, and the earliest ancestor prediction time is about 1984. The evolution rate of the B-lineage L segment was 7.44 × 10^−5^, and the earliest ancestor prediction time was about 1953 (excluding the ON812166.1, which is not on the same branch as other sequences and corresponds to a negative earliest ancestor time). The evolution rate of the F lineage L segment was 2.43 × 10^−4^, and the earliest ancestor prediction time was about 1984. The analysis of dissemination data is shown in Appendix A. From the data, it can be known that the F-lineage SFTSV only spreads between China and South Korea, with a greater possibility of transmission from South Korea to China. B-lineage SFTSV has spread among four countries: China, Japan, South Korea, and Thailand. There is a possibility of mutual transmission between China, South Korea, Japan, and Thailand. Thailand may have the possibility of unilateral transmission to Japan, while Japan may have the possibility of unilateral transmission to South Korea. Then, based on the BEAST results, spreaD3_v0.9.7.1rc was used to analyze the likelihood of propagation, and BAYESVNet > 3 was considered to have propagation paths. In addition, from the data, it could be known that the F-lineage SFTSV only spreads between China and South Korea, with a greater possibility of transmission from South Korea to China. China has a possibility of mutual transmission with South Korea, Japan, and Thailand. The propagation route is shown in the following Figure 7 (map source https://www.cnsknowall.com/#/HomePage).

## 4. Discussion

SFTSV was first discovered in 2009, and the virus was isolated in 2011. SFTSV spread to 27 provinces in China and it has also been confirmed in South Korea, Japan, Thailand, Vietnam, Myanmar, and Pakistan. We searched and integrated most of the SFTSV literature reports from 2009 to 2023. The reports of SFTSV have never evaporated, and have been showing a trend of slowly expanding. That consideration of the differences between the epidemic lineages in China and abroad might find the source of transmission faster for new outbreak areas and patients of SFTSV, thereby taking more effective defense measures.

Our study explored the SFTSV complete genome sequence uploaded in NCBI and divided its lineages according to the existing literature. The main lineage of the three SFTSV fragments found so far was the F lineage. China had uploaded the most SFTSV data, and the uploaded data mainly came from patients. Japan and South Korea had uploaded sectional SFTSV data, and most of the uploaded lineages belonged to the B lineage. There were many documents on the division of SFTSV lineages. We referred to the literature and NCBI, which may provide a clearer reference for the division of SFTSV lineages. However, it is still hoped that more authoritative institutions or the literature can provide a clearer definition of the division of SFTSV lineages.

The conservation of genes in different lineages of SFTSV may be one of the possible reasons for the difference in the spread of SFTSV at home and abroad. Based on the existing data, the nucleotide mutation frequency of the SFTSV F and B lineage was explored and analyzed. Taking the earliest uploaded sequence as a reference, it was concluded that the S and L fragments were relatively conservative overall, and the mutation rate of the M fragment was relatively high. The three fragments of the F and B lineage were analyzed separately. The mutation site protein prediction simulation analysis was performed on the different SFTSV fragments with changes in protein polarity to vividly display the protein mutation status. It was shown that there was no change in protein polarity caused by nucleotide mutation in the S fragment of the B lineage, which is considered to be the reason for the difference in transmission between the SFTSV B lineage and the F lineage. Considering that the virus spread to a new environment, the probability of virus mutation also changes, but the B lineage sequence of SFTSV was more popular abroad and more conservative. The reasons behind this still need to be carefully studied and investigated. Subsequently, a 3D protein model was constructed based on mutation analysis data. From the previous research literature, it can be found that certain amino acid mutations at certain sites had been studied and published for their effects on viruses. For example, protein mutations in the M fragments R624W and R962S altered the virus’s ability to fuse with cells [21,22]. The mutation at position 926 of the M fragment altered the growth rate of the virus. The protein mutation at position N1891K in the L fragment can affect the activity of viral polymerase and may even affect its function in genome transcription and viral replication [23]. Only a few protein mutation sites had been experimentally confirmed in this analysis, and the effects of most mutated amino acids had not yet been discovered. Among them, most of the confirmed mutations in the literature were mutation sites of the F lineage. Therefore, more researchers need to discover the differences between these two lineages, to provide basic information for SFTSV prevention across different lineages.

We can know that the appearance of SUMOylation, palmitoylation, and N-glycosylation, the protein-functional site mutation of prediction simulation analysis was performed on the different SFTSV fragments with changes in protein polarity to more vividly display the protein mutation status. It can lead to changes in the interaction between the virus and the host. A part of the difference between the two SFTSV lineages was reflected at the host level [24,25,26,27]. After analyzing the nucleotide and amino acid mutations, we used the online prediction website to deeply predict the protein property site of the protein-coding regions in different SFTSV lineages and know that the F lineage and B lineage of the S fragment did not have SUMOylation sites or palmitoylation sites. But there was an N-glycosylation site at the same amino acid site 17. It is speculated that the nucleotide sequence changes in the S fragment are not the main reason for the generation of its two lineages. The F lineage and S lineage of the M fragment did not have SUMOylation sites, but both had palmitoylation sites and N-glycosylation sites. At present, only a few differences in quantity can be found between the two lineages and more in-depth research may require a combination of experiments and theory.

Gene recombination is the main source of virus variation, and for segmented viruses, gene recombination can directly unfold the “exchange and recombination” of all fragments. However, there is currently a lack of feasible prediction models for these segmented viruses. Here, gene recombination can only predict the recombination within three fragments, and cannot determine whether recombination between fragments has occurred. Moreover, for this type of recombinant virus composed of multiple lineages, there is no definite standard. From the recombination within the fragments above, it can be seen that the M fragment of SFTSV is closely related to the B-lineage SFTSV, while the F lineage is mainly related to the L fragment. If the virus strains produced by the crossing of the F and B lineages were artificially rescued, it would be more helpful for studying the differences between the two lineages.

Based on the 50 sequences of the minimum S fragment B lineage obtained through lineage splitting, sequences with similar numbers were randomly selected from other segments and lineages for analysis and comparison of evolutionary rates. It was known that the SFTSV evolution rate of the B lineage popular abroad was generally greater than the F lineage. It may be one of the reasons why the SFTSV of the B lineage had a wider range of dissemination. Secondly, by analyzing SFTSV about the earliest ancestor prediction time of the F and B lineages, it was found that the S and L fragment’s earliest ancestors of the B lineage were both earlier than the F lineage, while the B lineage M fragment appeared later than those of the F lineage. Here, it is hypothesized that there was a possibility that the B-lineage SFTSV had a wider and faster spread in Southeast Asia compared to the F lineage due to some changes in the M segment. From Figure 7, it can also be seen that the spread range of the B lineage is larger and the prevalence region is more apparent. The recent literature explains this issue from the perspective of geographical branches, and it is true that SFTSV has two different transmission scenarios. The B lineage may not have been newly mutated in recent years, while the F lineage may have been present in ticks for a long time. A new perspective has been proposed in the literature that the prevalence of SFTSV depends on the adaptive mutation of a certain fragment to the host, resulting in a new lineage. This mutation does not appear in isolation in a single fragment, but rather in two fragments that correspond to each other, indicating that at least three fragments will have pairwise effects and generate new lineages [28]. Moreover, it was speculated that the three segments of SFTSV are not correlated one-to-one. In the above data analysis and the other literature, S and M fragments exhibited identity features (mutation analysis), and S and L fragments also exhibited identity features (population abundance, lineage division). It seems that there is a special connection channel among them. At present, more effective data cannot be obtained from prediction results between the two lineages, and unable to explain the order of the two lineages well. More SFTSV lineage data and prediction models are needed to analyze the formation time and order of lineages, as well as the locations where they are likely to spread next. It will provide significant help for preventing SFTSVs of different lineages in the future.

Our study attempted to analyze the reasons for the distinction in the epidemic range and transmission hosts of the two lineages by analyzing the SFTSV of the F and B lineages. Finally, it is found that the distinction between the two lineages should have little to do with protein site and the possibility of gene recombination, but may be related to the changes in amino acid properties caused by the nucleotide mutation of the virus and the evolution rate. The genetic data analysis of these two lineages will provide further experimental research basis data for the spread of B-lineage SFTSV and the study of other lineages. However, more specific and detailed research requires more F and B lineage sequences for verification and more in-depth research to better prove it. In addition, this article did not conduct individual gene analysis for all lineages (only partial gene analysis for all lineages), making it impossible to form a comparative analysis from the whole to the parts. Secondly, the analysis of segmented viruses can only be conducted separately, without a good process and method for correlation analysis. The three fragments of SFTSV nucleic acid uploaded in NCBI were not matched before joint analysis, and the correlation between the fragments was not high. These factors may affect the final results of experimental data and more data analysis models are needed to accurately determine in the future.

## 5. Conclusions

This study indicates that there are certain genetic differences between F- and B-lineage SFTSVs, but further data and analysis models are needed for further analysis. The F and B lineages of SFTSV are both spreading within China. Although there are differences in transmission areas, further in-depth and detailed research is needed to better prevent SFTSV in both lineages. We cannot only detect the S segment, as all three segments of SFTSV have a clear impact on propagation differences. The B lineage of SFTSV has a wider range of transmission and more countries. Countries in Asia need to be vigilant and improve the sensitivity and simplicity of detecting B-lineage SFTSV, with a focus on prioritizing the development of vaccines targeting the B lineage. But we cannot forget to monitor the F lineage. Once the F lineage spreads widely, it may be a completely new outbreak of the epidemic. Clarifying the differences between the two SFTSVs of the F and B lineages will have a significant impact on the prevalence and preventive treatment of SFTSVs.

## Figures and Tables

**Figure 1 microorganisms-13-00292-f001:**
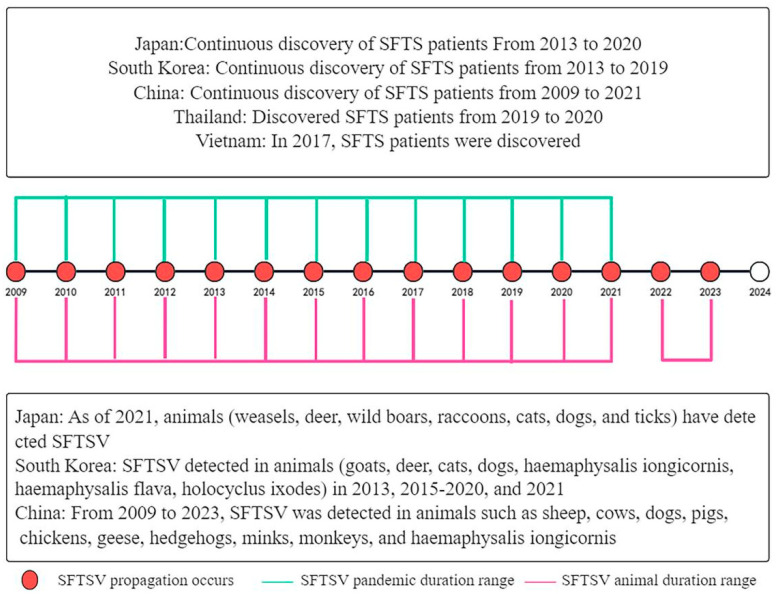
SFTSV epidemic timeline. By searching for the literature related to SFTSV, a timeline of prevalence was plotted. The upper part of the timeline shows the prevalence of SFTSV patients from different countries, while the lower part shows the detection of SFTSV in animals from different countries. Connecting lines at different years indicated the existence of temporal continuity, while unconnected lines indicated detection only within the current year.

**Figure 2 microorganisms-13-00292-f002:**
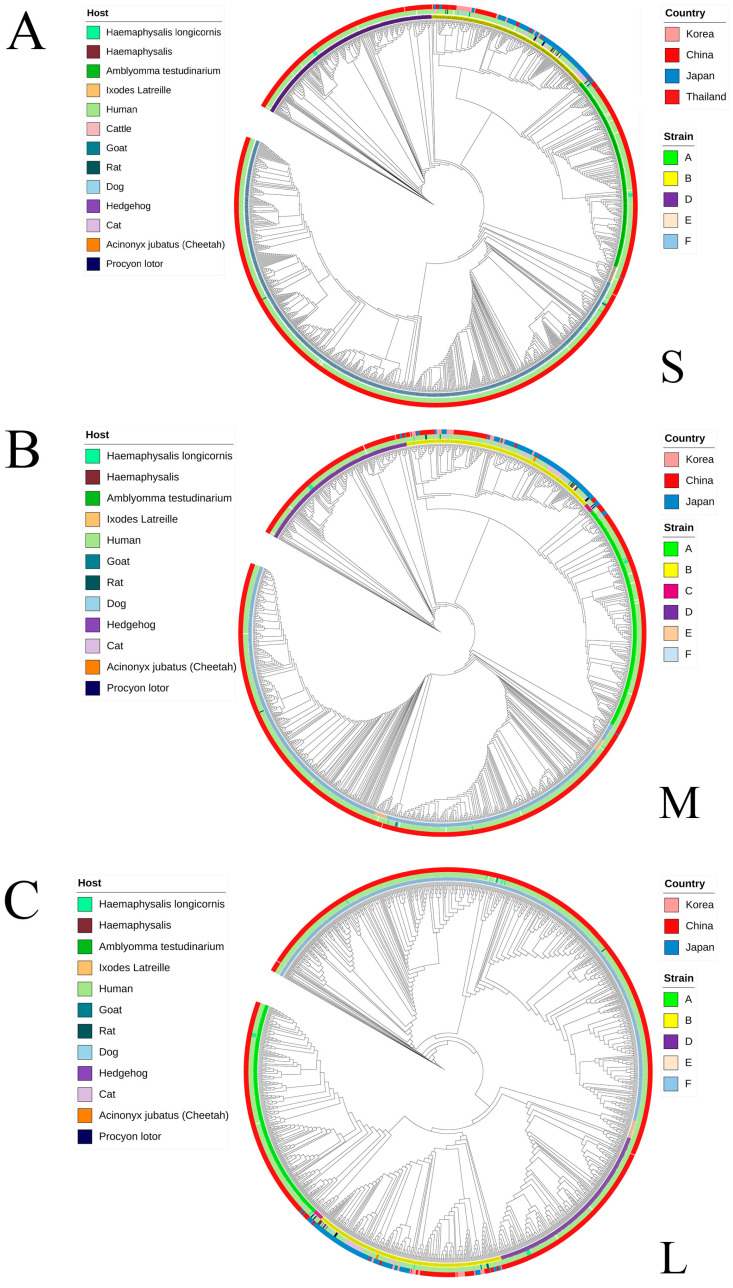
SFTSV ML tree for lineage division. (**A**) SFTSV S fragment ML tree. (**B**) SFTSV M fragment ML tree. (**C**) SFTSV L fragment ML tree. The color in the innermost circle represents the lineage, the middle of the circle represents the host, and the outermost circle represents the sample source of the country.

**Figure 3 microorganisms-13-00292-f003:**
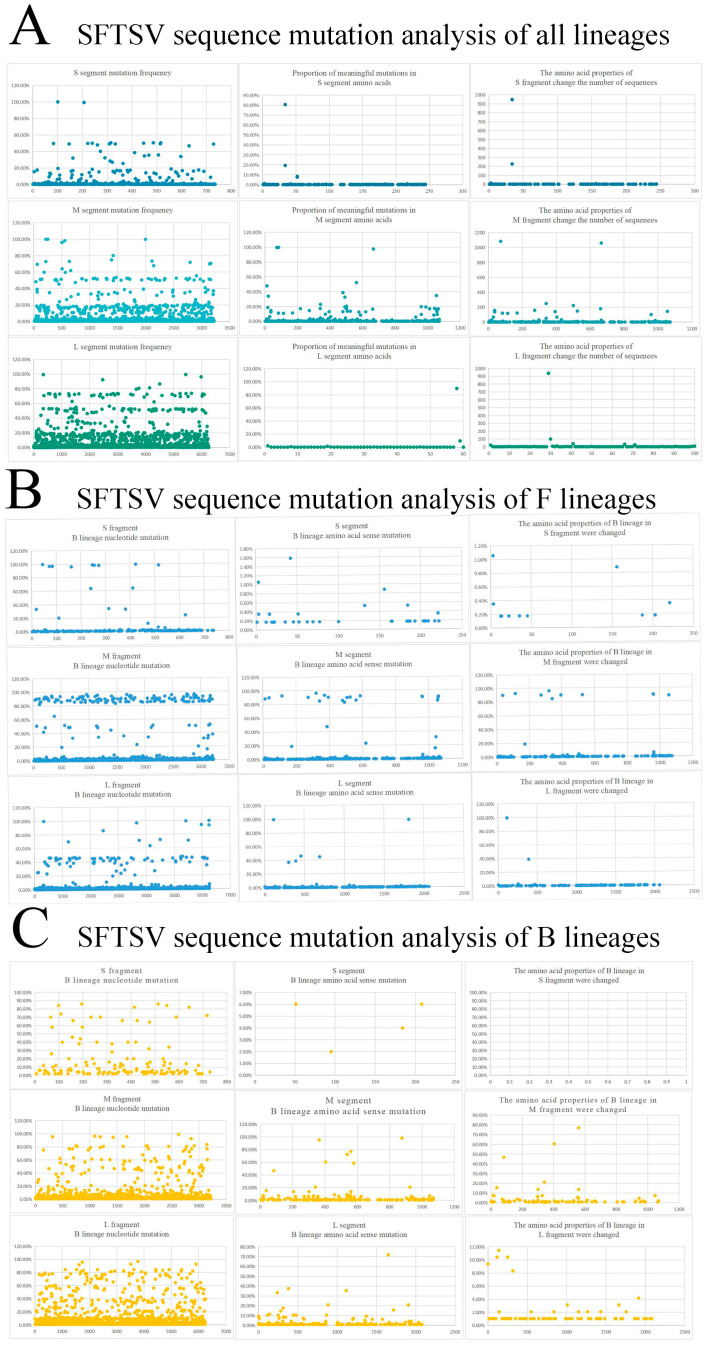
Scatter plot of SFTSV mutation frequency analysis. (**A**) Analysis of all sequence mutations in SFTSV. (**B**) Analysis of mutations in B-lineage SFTSV. (**C**) Analysis of mutations in F-lineage SFTSV.

**Figure 4 microorganisms-13-00292-f004:**
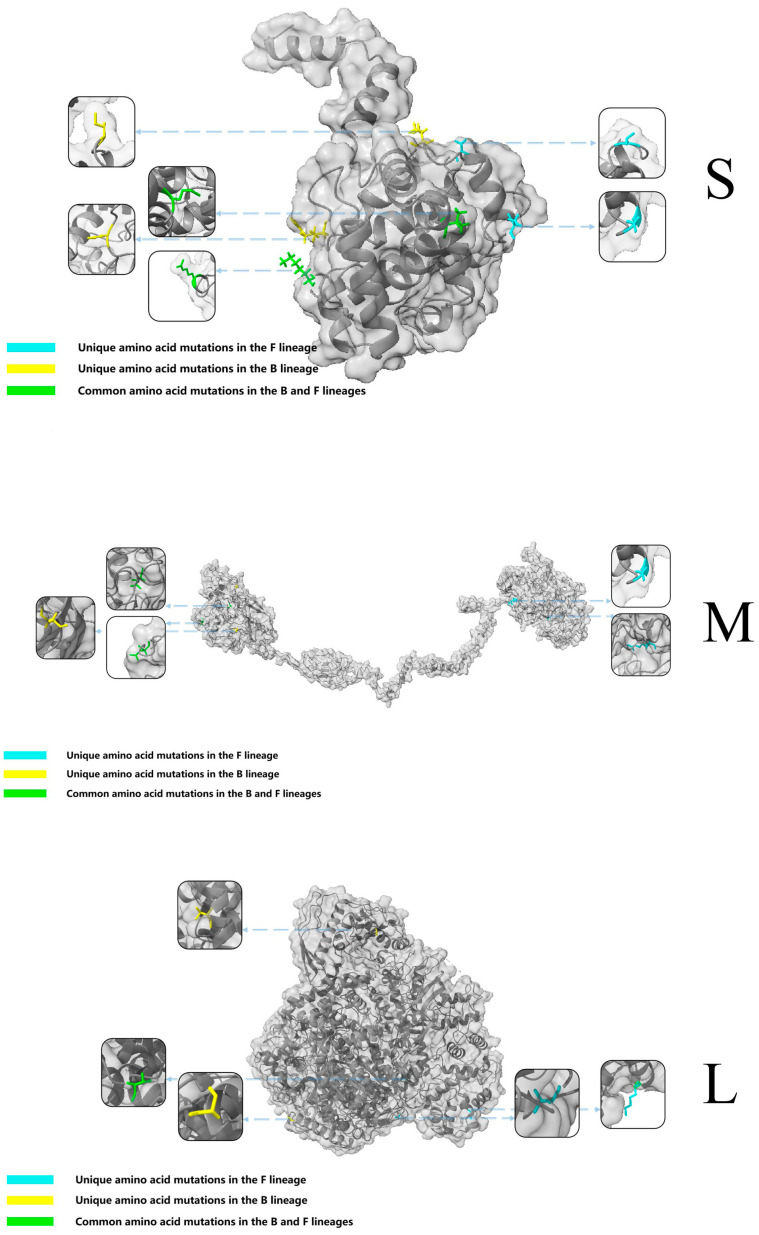
Structural simulation of mutant proteins in F and B Lineage SFTSV. Yellow represents unique mutations in the B lineage, blue represents unique mutations in the F lineage, and green represents common mutations in the two lineage.

**Figure 5 microorganisms-13-00292-f005:**
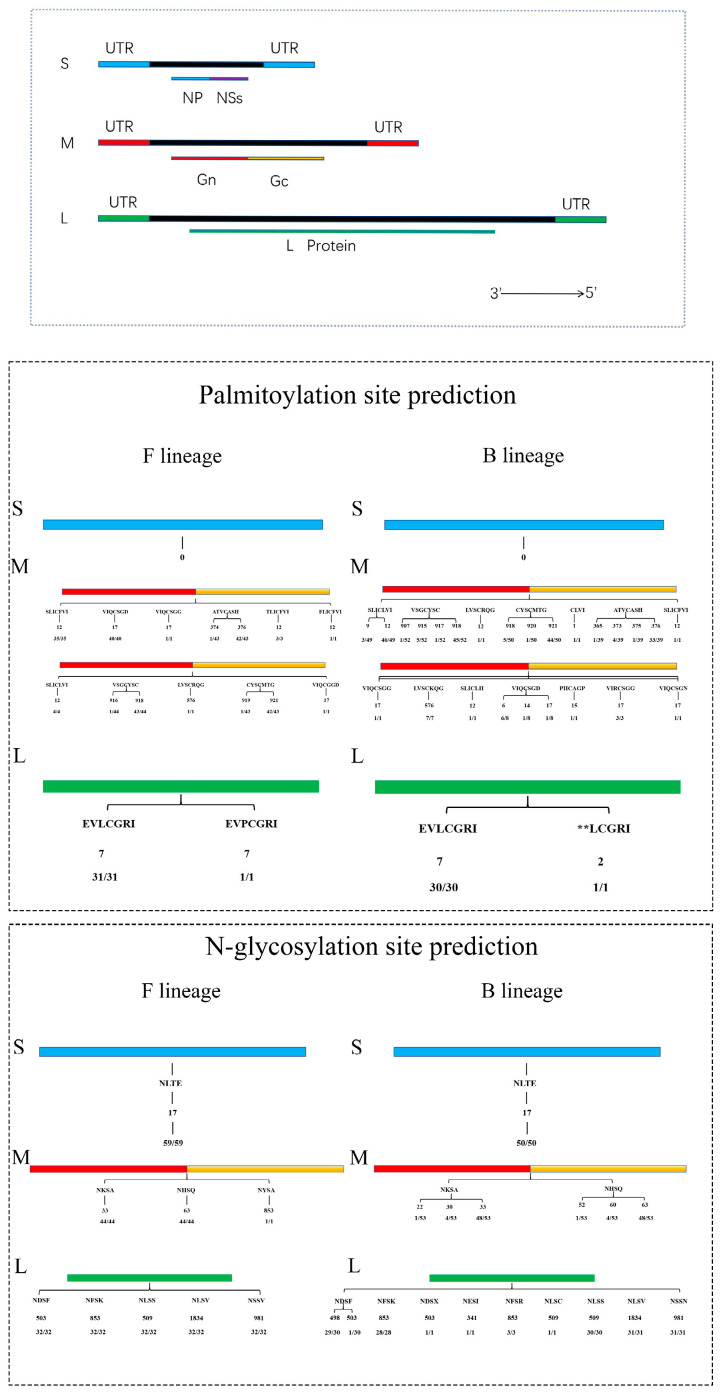
SFTSV functional site prediction. (**A**) Visualization of SFTSV fragments and encoded proteins, with fragment colors corresponding to protein functional site prediction. (**B**) Prediction of palmitoylation sites in F and B-lineage SFTSV (In order to standardize the format, they are all seven characters long, ** representing the absence of amino acids). (**C**) Prediction of N-glycosylation sites in F and B-lineage SFTSV.

**Figure 6 microorganisms-13-00292-f006:**
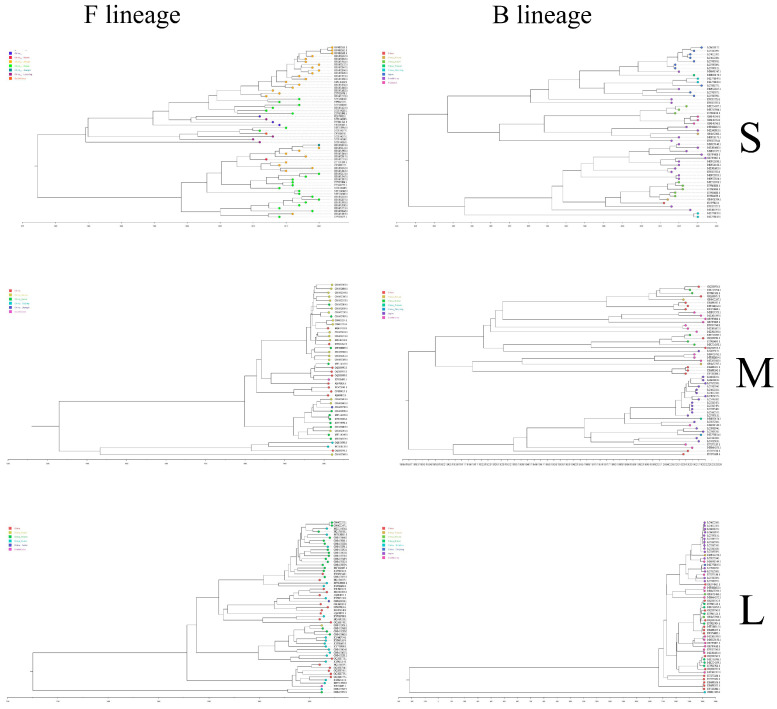
F and B lineages of SFTSV MCC trees. The MCC tree is visualized in Figtree, where the color of the tree branch endpoints corresponds to the country.

**Figure 7 microorganisms-13-00292-f007:**
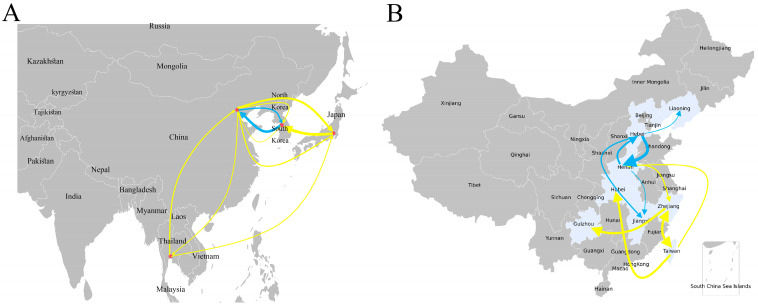
SFTSV F and B lineage spread prediction map. (**A**) SFTSV F and B lineage Asian spread prediction map. The red dot represents the capital of the country, the yellow arc line represents the spread pathway of B-lineage SFTSV, the blue arc line represents the spread pathway of F-lineage SFTSV, the arrow represents the direction of spread, and the thickness of the arc line represents the possibility of spread. (**B**) SFTSV F and B lineage China spread prediction map. The meaning represented by the arc line is the same as above. The light blue area shows the province used for spread analysis.

## Data Availability

The original contributions presented in this study are included in the article/Appendix A. Further inquiries can be directed to the corresponding authors.

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
