# Peer review of "Analysis of Gene Differences Between F and B Epidemic Lineages of Bandavirus Dabieense"

_microorganisms, 2025, doi:10.3390/microorganisms13020292_

Round 1

Reviewer 1 Report

Comments and Suggestions for Authors

Adjust sections and citation to journal style.

The introduction should be reorganized to clearly outline the purpose of the study and justify the choice of the two epidemic lines (F and B).

Include details on how sequences were selected and how databases were checked to ensure the representativeness of the F and B lines.

Expand the discussion on how specific mutations in viral proteins affect virus-host interaction, transmission, and virulence.

Compare the study results to previous research on the evolution of similar genotypes in SFTSV or vector-borne viruses.

Acknowledge that the results are based on bioinformatics predictions and that additional functional studies are required to confirm the observed differences between the F and B lines.

The conclusions should include specific recommendations for epidemiological surveillance or the design of control measures based on the genetic differences between the F and B lines.

Author Response

Note: All modifications in the literature are marked with a red background, and responses are in red.

Reviewer #1

Comments 1:Adjust sections and citation to journal style.

Response 1: Thank you for pointing this out. Therefore, l have adjusted the abstract and citation style. Including modifying the format of the abstract, adjusting the superscripts of citations, and adding DOI to references.  (Change the line number of the content to 533-599)

Comments 2: The introduction should be reorganized to clearly outline the purpose of the study and justify the choice of the two epidemic lines (F and B).

Response 2: Thank you for your suggestion, so l have revised the introduction and added the research purpose of this article, as well as the reasons and rationale for selecting DBV B and F lineages for gene difference analysis. Starting from the uniqueness of the B lineage in SFTSV public data, I compared the genetic differences between the F lineage with the most detected data. I used various prediction methods and software to determine the differences between the two lineages. (Change the line number of the content to 50-68)

Comments 3: Include details on how sequences were selected and how databases were checked to ensure the representativeness of the F and B lines.

Response 3: Thank you for pointing this out. Firstly, based on the evolutionary tree diagram 1, it was found that the sequence number of DBV's S fragment B lineage was the smallest, at 50. Therefore, this was used as the screening quantity standard to remove sequences with high similarity (host, time/Animal host as the first screening principle, year difference as the second screening principle). Finally, it was found that some of the analyzed data had clear lineage divisions on NCBI, which was consistent with this article. Specific analysis sequence selection is added in the method. (Change the line number of the content to 91-97)

Comments 4:Expand the discussion on how specific mutations in viral proteins affect virus-host interaction, transmission, and virulence.

Response 4: Thank you for pointing this out. l agreewith this comment. I have reorganized the discussion on virus mutations in the article and added the reasons for the changes in the mechanism of action after mutations, indicating that mutations do indeed alter the transmission, virulence, and interactions of viruses. (Change the line number of the content to 406-418)

Comments 5:Compare the study results to previous research on the evolution of similar genotypes in SFTSV or vector-borne viruses.

Response 5: I compared the final spread analysis of the B and F lineages with a previous SFTS analysis literature, which indirectly confirms that SFTS does indeed have two lineages with genetic and epidemiological differences. (Change the line number of the content to 458-466)

Comments 6: The conclusions should include specific recommendations for epidemiological surveillance or the design of control measures based on the genetic differences between the F and B lines.

Response 6: Thank you for your comment. I think the revised conclusion includes possible preventive measures, including the need to correct the sensitivity of the B lineage SFTSV detection method, vaccine development for the B lineage, continuous monitoring of the transmission range of the F lineage, and so on. But personally, I believe that if SFTV has aerosol transmission, developing targeted vaccines is more important here. (Change the line number of the content to 500-506)

Reviewer 2 Report

Comments and Suggestions for Authors

The reviewed study presents robust data on the genetic distinctions and epidemiological behavior of the F and B lineages of SFTSV. It successfully integrates phylogenetic analysis and genetic mutation data to establish lineage-specific transmission and mutation characteristics, effectively highlighting their broader implications for virus spread and host interaction. The research employs comprehensive statistical and modeling methods, including advanced software tools, to ensure rigorous analysis of protein functional sites and evolutionary rates. Its meticulous data collection from multiple countries ensures a well-rounded understanding of SFTSV's prevalence and evolutionary trajectory. The insights provided into lineage-specific traits offer valuable perspectives for future research and preventive strategies against SFTSV transmission.

The study requires improvement in clarifying and standardizing lineage naming conventions to ensure consistency with existing literature and enhance interpretability. A stronger emphasis on experimental validation of predicted mutation effects and functional site changes is necessary to substantiate theoretical findings and provide more actionable insights. Additionally, the research would benefit from integrating comprehensive gene interaction and recombination analyses to better understand the interplay between different genome fragments and their collective impact on SFTSV transmission and evolution.

Items to correct/improve:

There is a limited exploration of gene interactions across SFTSV fragments, hindering a comprehensive understanding of their combined impact on lineage behavior

The manuscript has a lack of direct experimental validation for several proposed mutation effects and protein functional site predictions

There is insufficient focus on correlating recombination events with evolutionary traits of the F and B lineages

The paper contains ambiguities in the lineage naming conventions and inconsistencies with existing literature, which could affect interpretation

There is limited geographic scope of some analyses, restricting broader generalization of the findings

The manuscript has a lack of detailed individual gene analysis for a more nuanced understanding of lineage-specific mutations

There is a reliance on fragmented data for joint analysis without effectively addressing inter-fragment relationships

The reviewer sees an absence of a clear methodology for linking nucleotide mutations to changes in virus-host interactions

The minimal discussion on the implications of the observed transmission patterns for public health interventions can be expanded.

The paper could benefit from more robust and diverse prediction models to enhance the accuracy of evolutionary and spread analyses.

Author Response

Note: All modifications in the literature are marked with a red background.

Reviewer #2

The reviewed study presents robust data on the genetic distinctions and epidemiological behavior of the F and B lineages of SFTSV. It successfully integrates phylogenetic analysis and genetic mutation data to establish lineage-specific transmission and mutation characteristics, effectively highlighting their broader implications for virus spread and host interaction. The research employs comprehensive statistical and modeling methods, including advanced software tools, to ensure rigorous analysis of protein functional sites and evolutionary rates. Its meticulous data collection from multiple countries ensures a well-rounded understanding of SFTSV's prevalence and evolutionary trajectory. The insights provided into lineage-specific traits offer valuable perspectives for future research and preventive strategies against SFTSV transmission.

The study requires improvement in clarifying and standardizing lineage naming conventions to ensure consistency with existing literature and enhance interpretability. A stronger emphasis on experimental validation of predicted mutation effects and functional site changes is necessary to substantiate theoretical findings and provide more actionable insights. Additionally, the research would benefit from integrating comprehensive gene interaction and recombination analyses to better understand the interplay between different genome fragments and their collective impact on SFTSV transmission and evolution.

Comment 1: The exploration of gene interactions between SFTSV fragments is limited, hindering a comprehensive understanding of their combined impact on lineage behavior.

Response 1: Thank you for pointing it out. I acknowledge this comment. However, there is currently no systematic analysis method in the literature for the interactions between different fragments of SFTS. Some literature associates different fragments with consistent upload sources (suggesting that these fragments belong to the same virus) and attempts to analyze them. I think this method has too much error and there are too few sequences in the B lineage. If some fragments are deleted, the subsequent SFTS analysis of the B and F lineages may fail. One important pathway for gene interaction between fragments is gene recombination, so I have redefined the relationship between gene recombination and the F and B lineages and added an extension of gene recombination in the discussion. (Change the line number of the content to 314-322, 435-445)

Comments 2:The manuscript has a lack of direct experimental validation for several proposed mutation effects and protein functional site predictions.

Response 2: Thank you for pointing it out. I acknowledge this comment. However, there is currently no systematic analysis method in the literature for the interactions between different fragments of SFTS. Some literature associates different fragments with consistent upload sources (suggesting that these fragments belong to the same virus) and attempts to analyze them. I think this method has too much error and there are too few sequences in the B lineage. If some fragments are deleted, the subsequent SFTS analysis of the B and F lineages may fail. Due to the limitations of available data and the lack of predictive models, it is difficult to achieve the expected results. But we will continue to learn and try more methods in future research. In addition, One important pathway for gene interaction between fragments is gene recombination, so I have redefined the relationship between gene recombination and the F and B lineages and added an extension of gene recombination in the discussion. (Change the line number of the content to 314-322, 435-445)

Comments 3:There is insufficient focus on correlating recombination events with evolutionary traits of the F and B lineages.

Response 3: Thank you for pointing this out. I agree with this comment. I have re examined the intra segmental recombination of SFTSV, deepened the correlation between gene recombination and the B and F lineages, and re discussed the current situation. The gene recombination phenomenon of SFTSV not only includes intra fragment recombination, but also inter fragment recombination. However, current analysis software can only analyze intra fragment recombination, and inter fragment recombination can only be processed and analyzed bit by bit. Therefore, this article does not discuss the situation of inter fragment recombination. (Change the line numbers of the content to 314-322, 435-445)

Comment 4: There is ambiguity in the naming convention of the genealogy in this paper, which is inconsistent with existing literature and may affect the interpretation.

Response 4: Thank you for pointing this out. Allow me to explain here: There are various classification forms in the current research literature on SFTSV, including A-F lineage, separate B1, B2, B3 lineage, C1, C2, C3, J1, J2, J3 lineage named after countries, and newly designed lineages based on experimental requirements. It is precisely because of this chaotic classification standard that I advocate for the adoption of more standardized virus classification methods as soon as possible in the discussion. The A-F lineage I have chosen has a relatively high acceptance rate, and this literature also includes a relatively large number of lineage divisions. 

Yongfeng, Fu,Shibo, Li,Zhao, Zhang et al. Phylogeographic analysis of severe fever with thrombocytopenia syndrome virus from Zhoushan Islands, China: implication for transmission across the ocean. Sci Rep, 2016, 6: 0.

Comments 5: The limited geographical scope of some analyses restricts the broader generalization of research results.

Response 5: Thank you for pointing this out. Allow me to explain here: the data obtained from the analysis is sourced from SFTSV data uploaded by various countries, and the geographical limitations depend on the country of the data source, which cannot directly expand the geographical scope of the analysis. However, SFTSV has similar viruses appearing in other countries, but we have not included similar viruses in our unified analysis, which is also our shortcoming. Considering that although these viruses belong to the same genus, they are not the same virus, they were ultimately removed after conditional screening. Furthermore, Due to the limitations of available data, it is difficult to achieve the expected geographical prediction location. But we will search for more ideal prediction models and obtain more effective results in future research.

Comments 6: The manuscript has a lack of detailed individual gene analysis for a more nuanced understanding of lineage-specific mutations.

Response 6: Thank you for pointing this out. Allow me to explain here: The purpose of this study is to explore the reasons for the differences in transmission between the two lineages of SFTSV viruses, and therefore attempt to analyze and explore them from a genetic sequence perspective. Thus, the sequence with the earliest data upload time was selected as the reference. Therefore, we did not analysis our own discovery of SFTSV sequences and focused more on mining public data. Or rather, what you mean is that the article lacks analysis of protein mutations in the specific sequences of lineage B and F, which has been rephrased and added to the discussion. (Change the line number of the content to 406-418)

Comments 7: There is a reliance on fragmented data for joint analysis without effectively addressing inter-fragment relationships.

Response 7: Thank you for pointing this out. Allow me to explain here: At present, there is no literature research on the relationship between SFTSV virus fragments, nor is there any research result on similar segmented viruses of the same type. I can only add relevant literature content in the discussion to expand and extend it. (Change the line number of the content to 458-466)

Comments 8: The reviewer sees an absence of a clear methodology for linking nucleotide mutations to changes in virus-host interactions.

Response 8: Thank you for pointing this out. Please allow me to explain here: there must be some kind of correlation between virus host interaction and nucleotides, which we are currently unable to explain well due to available data and predictive models. This requires our further research and experimentation. Therefore, I believe it is helpful to discuss the reasons for nucleotide mutations in the B and F lineages of SFTSV. In addition, nucleotide mutation analysis can be followed by amino acid analysis, which can further enrich the impact of mutations.

Comments 9: The minimal discussion on the implications of the observed transmission patterns for public health interventions can be expanded.

Response 9: Thank you for pointing this out. I made adjustments in the conclusion and added appropriate preventive and targeted measures. This includes updating the priority detection methods for the B lineage in both lineages, developing vaccines, and increasing monitoring efforts for the transmission range of the F lineage. (Change the line number of the content to 500-506)

Comments 10: The paper could benefit from more robust and diverse prediction models to enhance the accuracy of evolutionary and spread analyses.

Response 10: Thank you for pointing this out. Because the amount of data that needs to be processed is enormous, coupled with the software and methods that need to be learned. I have also tried building other prediction models before, such as selection pressure, codon preference, protein domain prediction, and so on. However, the final prediction result cannot be obtained, so it was not included in the literature. We will try to learn more data prediction models in future research to obtain more effective prediction results

Reviewer 3 Report

Comments and Suggestions for Authors

The study presents a detailed phylogenetic analysis of a relatively not much studied virus that infects mainly human, but has been also reported in animals. The analysis is very well performed, supported by several methodologies. However, the main inferences such as the correlation with the different countries and a second less discussed finding, the correlation between animals and human, would be better benefited by a Network analysis, which I suggest to be carried out by the authors.

Authors are also recommended to add at least one more sentence in the abstract background to provide more insights for the SFTSV. Additionally the abbreviation has to be explained in the first time mentioned. Also, in the Methods of the abstract the authors have to mention the molecular technique they performed.

Line 103 what animal species? Please be specific

Lines 101-112. It would be better to present all this info in a Table.

Figures 3 and 4 should be enlarged. It is hard to see them

Generally, the study worths publication, and apart form my minor suggestions above, I only recommend a Network analysis that would improve the quality of the manuscript.

Author Response

Note: All modifications in the literature are marked with a red background.

Reviewer #3

Comments 1:The study presents a detailed phylogenetic analysis of a relatively not much studied virus that infects mainly human, but has been also reported in animals. The analysis is very well performed, supported by several methodologies. However, the main inferences such as the correlation with the different countries and a second less discussed finding, the correlation between animals and human, would be better benefited by a Network analysis, which I suggest to be carried out by the authors.

Response 1: Thank you for pointing this out. I have been trying to construct a network analysis graph using sequences from two lineages in the past few days. However, neither the correlation parameters between nucleotide sequences nor the predicted data of lineage transmission locations can be used to construct the network analysis graph. The network analysis graph requires relevant data mainly obtained from metagenomic sequencing. There may be other methods to construct network analysis graphs using online data,Due to the lack of available data, we will strengthen our learning and exploration in this area in future research.

Comments 2:Authors are also recommended to add at least one more sentence in the abstract background to provide more insights for the SFTSV. Additionally the abbreviation has to be explained in the first time mentioned. Also, in the Methods of the abstract the authors have to mention the molecular technique they performed.

Response 2: Thank you for pointing this out. l agreewith this comment. I reorganized the abstract according to the format of the submitted literature and added this sentence at the end. We have added explanations to the abbreviations again and incorporated molecular technology.  (Change the line number of the content to 26, 32-34, 39-52)

Comments 3:Line 103 what animal species? Please be specific.

Response 3: Thank you for pointing this out. I agree with this comment. So a note was added after the last sentence of the paragraph, and more detailed content can be found in Supplementary Table 2. (Change the line number of the content to 119-122)

Comments 4:Lines 101-112. It would be better to present all this info in a Table.

Response 4: Thank you for pointing this out. I agree with this comment. I have reorganized the data processed by the lineage and added a data description at the end of Supplementary Table 2.

Comments 5: Figures 3 and 4 should be enlarged. It is hard to see them.

Response 5: I have re-inserted the enlarged images of Figure 3 and Figure 4, but did not change the frame rate of the images because the frame rate requirement is 300dpi. I hope it can be seen clearly. If you need clearer images, please reply and I will add images with better frame rates. (Change the line number of the content to 270-279)

Comments 6: Generally, the study worths publication, and apart form my minor suggestions above, I only recommend a Network analysis that would improve the quality of the manuscript.

Response 6: Thank you very much for your suggestion. However, our article still uses publicly available data as raw data, and simple sequence information seems insufficient to complete the network analysis graph. But I also tried here, using SFTSV F and B lineage sequences to align identity with the earliest sequence, and constructed a host difference grid map of M fragments based on identity differences. I am not sure if it can be used. Attached with the file. From the graph, it can be seen that human sequences can still cover animal sequences, but this may be the result of a significant difference in the amount of data between the two. It may not be possible to complete a good image at the moment.

Round 2

Reviewer 2 Report

Comments and Suggestions for Authors

The reviewer comments were properly addressed. The paper can now be recommended for publication.